# ¿Son adecuados los Algoritmos Bio-Inspirados Binarios Recientes para Selección de Características?

**Francisco Javier Rodríguez Díaz**
Departamento de Ciencias de la Computación
Universidad de Granada
fjrodriguez@decsai.ugr.es

**Miguel García López**
Universidad de Granada
migue8gl@correo.ugr.es

**Daniel Molina Cabrera**
Departamento de Ciencias de la Computación
Instituto Andaluz Interuniversitario de de Data Science and Artificial Intelligence
Universidad de Granada
dmolina@decsai.ugr.es

## Abstract

Los algoritmos bioinspirados han ganado mucha popularidad en los últimos años, produciéndose un gran número de propuestas cada año. Sin embargo, estudios recientes han cuestionado el grado de novedad de gran parte de ellos. Y no solo eso, también la mejora experimental que pueden ofrecer frente a otros algoritmos más consolidados en la literatura. Aunque la mayoría de esos algoritmos recientes se han diseñado inicialmente para optimización continua, también existen versiones binarias para abordar problemas como la selección de características. Dado que se duda de las ventajas competitivas de los versiones originales, dicha duda también se puede extender hacia estas versiones binarias, sobre si realmente superan a los métodos metaheurísticos más consolidados en la literatura. Para responder a esta pregunta, este trabajo compara la eficiencia de algoritmos bio-inspirados binarios recientes con otros más tradicionales para la selección de características en distintos problemas de clasificación. Los resultados indican que la mayoría de estos nuevos algoritmos binarios no mostraron un rendimiento significativamente mejor en comparación con otras técnicas más tradicionales, como los algoritmos genéticos o los sistemas de partículas, y en el caso más competitivo, lo conseguía a costa de un mayor coste computacional. Por tanto, estos recientes algoritmos binarios no aportan ventajas prácticas sobre los métodos más clásicos. Esperamos que las conclusiones de este análisis ayuden a los investigadores a tomar decisiones fundadas a la hora de abordar la selección de características usando metaheurísticas, fuera de modas, contribuyendo así a un avance real, y más sostenible, de la inteligencia artificial.

## 1 Introducción

La computación Bio-inspirada es un area muy destacada dentro del campo de las metaheuristicas [Yang et al., 2013] que toma inspiración en procesos biológicos y de comportamiento de especies biológicas o de fenómenos naturales para diseñar algoritmos de optimización. El ejemplo más representativo son los Algoritmos Evolutivos, EA [Baeck et al., 1997], que han demostrado una gran habilidad para obtener soluciones de gran calidad en dominios de búsqueda muy complejos, desde problemas de ingeniería hasta redes profundas [Zhan et al., 2022, Martinez et al., 2021].

Dentro del aprendizaje automático, uno de los preprocesamientos más utilizados y con gran influencia es la selección de características (SC). La SC se puede definir como la identificación de atributos irrelevantes, o con ruido, de un conjunto de datos. La SC es relevante porque si se eliminan dichos atributos del proceso de aprendizaje, no solo mejora el tiempo de aprendizaje y clasificación, también el acierto de los modelos [Chandrashekar and Sahin, 2014]. Al tratarse de un problema NP-completo el uso de metaheurísticas es una buena estrategia en problemas con muchos atributos [Al-Sahaf et al., 2019].

Cada año el número de propuestas bio-inspiradas aumenta [Molina et al., 2020], y aunque la mayoría de estos algoritmos son para optimización de parámetros continuos, de muchos también se han desarrollado versiones binarias para SC. Sin embargo, varios investigadores han puesto en duda la conveniencia de muchos algoritmos, tanto a nivel de innovación [Sorensen, 2015], como desde su punto de vista práctico a nivel experimental [Tzanetos and Dounias, 2021]. Esto produce la paradoja de que el *éxito* en propuestas sea contraproducente, ya que muchas de esas propuestas no solo no aportan, sino que restan al introducir ruido [Molina et al., 2025].

Algunos estudios recientes extienden estas dudas a los algoritmos bioinspirados más populares [Camacho Villalón et al., 2020], encontrando sesgos significativos en el comportamiento de búsqueda de muchos de ellos [Kudela, 2023]. Sin embargo, aunque las versiones continuas han sido ampliamente estudiadas, sus correspondientes binarias, al ser menos populares, han recibido menos atención. La mayoría de los estudios se han centrado en comparar sólo unos pocos algoritmos bioinspirados binarios modernos [Dokeroglu et al., 2022], y sólo recientemente han empezado a surgir estudios más completos [Rostami et al., 2021].

El objetivo de este trabajo es analizar el comportamiento de versiones binarias de algoritmos bioinspirados recientes para SC comparándolos con otras metaheurísticas para responder la siguiente pregunta de investigación: *¿Las versiones binarias de los recientes algoritmos bioinspirados mejoran a otros metaheurísticas para selección de características?* Para abordar esta pregunta, evaluamos doce algoritmos, incluyendo seis versiones binarias recientes de algoritmos bioinspirados, y otras seis metaheurísticas establecidas, y los aplicamos para realizar una SC de distintos problemas de clasificación. Una vez aplicada la SC, aplicamos dicha selección sobre el conjunto de test, evaluando los resultados obtenidos, tanto en acierto como en reducción de características, y en tiempo computacional requerido. Los resultados muestran que, en la mayoría de los casos, los algoritmos más clásicos superan a los modernos, respondiendo así a nuestra pregunta.

Este artículo se estructura de la siguiente manera. En la Sección 2, describimos brevemente los algoritmos bioinspirados comparados. En la Sección 3, detallamos el marco experimental. En la Sección 4, presentamos los resultados obtenidos y proporcionamos un análisis de los mismos. Finalmente, en la Sección 5, resumimos las principales conclusiones obtenidas.

## 2 Algoritmos Bioinspirados Binarios

En esta sección describiremos muy brevemente los distintos algoritmos modernos utilizados en el estudio, junto con ejemplos de su uso en aplicaciones reales, y la versión binaria utilizada.

El algoritmo *Grey Wolf Optimization* (GWO) es un método inspirado en la estrategia de caza de los lobos grises, donde la jerarquía de la manada se divide en lobos alfa (mejor solución), beta, delta y omega [Mirjalili et al., 2014]. GWO ha sido aplicado en diversos problemas reales [Panda and Das, 2019] y su versión binaria (BGWO) fue diseñada para selección de características [Emary et al., 2016, Al-Tashi et al., 2019].

El *Whale Optimization Algorithm* (WOA) es una técnica metaheurística inspirada en la estrategia de caza de las ballenas jorobadas, en particular su método de alimentación mediante redes de burbujas [Mirjalili and Lewis, 2016]. Simula su comportamiento a través de tres fases: el cercado de la presa, el movimiento en espiral y la exploración aleatoria. Ha sido utilizado en aplicaciones como seguridad [Eid et al., 2024]. Recientemente, se ha desarrollado una versión binaria (BWOA) para selección de características [Hussien et al., 2019].

El *Firefly Algorithm* (FA) es un algoritmo inspirado en el comportamiento de las luciérnagas [Yang, 2009, 2014]. Se basa en la atracción bioluminiscente, donde las luciérnagas más brillantes representan mejores soluciones y atraen a las demás en función de su intensidad, que disminuye con la distancia. Existe también una versión binaria (BFA)[Sayed et al., 2018, Zhang et al., 2017] y tanto FA como BFA

han sido aplicados con éxito en áreas como procesamiento de imágenes, planificación y optimización industrial [Kumar and Kumar, 2021].

El *Bat Algorithm* (BA) es una técnica metaheurística inspirada en la ecolocalización de los murciélagos [Yang, 2010]. Simula cómo los murciélagos emiten pulsos ultrasónicos y ajustan su frecuencia, velocidad y tasa de emisión para localizar presas u obstáculos. Los algoritmos inspirados en murciélagos han demostrado ser efectivos en diversas aplicaciones industriales [Dao and Nguyen, 2024]. En la selección de características se aplica una versión binaria (BBA) [Nakamura et al., 2012].

El *Dragonfly Algorithm* (DA) es un método de optimización inspirado en el comportamiento de agrupamiento de las libélulas en la naturaleza [Mirjalili, 2016]. Imita las fases de agrupamiento observadas durante la evasión de depredadores, la migración y la caza. Los algoritmos inspirados en las libélulas han demostrado ser efectivos para aplicaciones del mundo real [Rahman et al., 2023]. Existe una variante (BDA) para selección de características [Mafarja et al., 2018].

El *Grasshopper Optimization Algorithm* (GOA) [Saremi et al., 2017] imita el comportamiento de agrupamiento de los saltamontes en sus fases de ninfa y edad adulta, modelando sus interacciones sociales de los saltamontes. GOA ha sido aplicado con éxito a problemas complejos [Saremi et al., 2017] y su versión binaria (BGOA) a selección de características [Saremi et al., 2017].

## 3   Marco experimental

En esta sección, describimos el marco experimental utilizado en nuestro estudio. En primer lugar, ofrecemos una breve descripción de los algoritmos comparados y los conjuntos de datos analizados. A continuación, presentamos la configuración de entrenamiento, haciendo hincapié en los parámetros del algoritmo y las condiciones experimentales.

Este trabajo evalúa el rendimiento de varios algoritmos bioinspirados binarios recientemente propuestos e introducidos en la Sección 2:

- Binary Bat Algorithm (BBA) [Nakamura et al., 2012].
- Binary Dragonfly Algorithm (BDA) [Mafarja et al., 2018].
- Binary Firefly Optimization Algorithm (BFA) [Zhang et al., 2017].
- Binary Grasshopper Optimization Algorithm (BGOA) [Mafarja et al., 2019] .
- Binary Grey Wolf Algorithm (BGWO) [Al-Tashi et al., 2019].
- Binary Whale Optimizer Algorithm (BWOA) [Hussien et al., 2019].

Los algoritmos anteriores van a ser comparados con algoritmos establecidos y ampliamente utilizados, aunque las descripciones de estos métodos de referencia quedan excluidas por limitaciones de espacio. Entre estos algoritmos de referencia se incluyen los siguientes:

- Artificial Bee Colony (ABC) [Karaboga, 2005].
- Binary Cuckoo Search (BCS) [Rodrigues et al., 2013].
- Binary Differential Evolution (BDE), este algoritmo discretiza las soluciones mediante la función sigmoidea como se explica en [Mirjalili and Lewis, 2013], que es utilizado por la mayoría de algoritmos bioinspirados recientes.
- Binary Particle Swarm Optimization (BPSO) [Kennedy and Eberhart, 1997].
- Cuckoo Search (CS) [Rodrigues et al., 2013].
- Genetic Algorithm (GA) [Mitchell, 1998].

Hemos seleccionado diversos conjuntos de datos ampliamente utilizados de la literatura, que cubren una amplia gama de características e instancias para evaluar el rendimiento a través de diversos escenarios del mundo real. Sus características principales se resumen en la Tabla 1.

Los parámetros para cada algoritmo se detallan en la Tabla 2, utilizando principalmente las recomendaciones de los autores, y con parámetros del GA seleccionados empíricamente. Para WOA, se utilizaron valores por defecto de una implementación secundaria [Li et al., 2019], ya que no se proporcionaron valores definitivos en trabajos anteriores. DA y GWO no requieren ajuste de parámetros.

Table 1: Conjuntos de Datos Ordenados por Número de Características

| Conjunto de datos | #Instancias | #Características | Área |
|---|---|---|---|
| sonar | 207 | 60 | Biología |
| spambase-460 | 459 | 54 | Ciencias de la Computación |
| spectf-heart | 348 | 44 | Medicina |
| waveform5000 | 5000 | 40 | Física |
| ionosphere | 350 | 34 | Predicción del tiempo |
| dermatology | 366 | 34 | Medicina |
| wdbc | 568 | 29 | Medicina |
| parkinsons | 200 | 22 | Medicina |
| zoo | 101 | 18 | Biología |
| wine | 182 | 13 | Enología |
| breast-cancer | 286 | 9 | Medicina |
| diabetes | 768 | 8 | Medicina |
| yeast | 1483 | 8 | Biología |
| ecoli | 336 | 7 | Biología |
| iris | 149 | 4 | Biología |

Todos los algoritmos se inicializaron con el mismo tamaño de población para garantizar una adecuada comparación de resultados. Las implementaciones y los resultados están disponibles en GitHub[1].

Los experimentos se realizaron en un clúster con diferentes nodos. Cada nodo tiene un Intel Core i7 y 24 GB de RAM. El ordenador ejecutaba una versión de 64 bits del sistema operativo Ubuntu 22.04 LTS. El código C++ se compiló y ejecutó utilizando el compilador g++ versión 11.4.

Table 2: Parameters of the different optimization algorithms

| Algoritmo | Parámetros |
|---|---|
| Compartido | Tamaño de la Población: 60 |
| GOA | $c_{min}$: 0.00001 $c_{max}$: 1 F: 0.5 L: 1.5 |
| WOA | Spiral parameter: 1 |
| ABC | Employed bee: 3 Onlooker bee: 3 Limit: 3 |
| BA | $\alpha$: 0.9 $\gamma$: 0.9 $f_{min}$: 0 $f_{max}$: 2 |
| PSO | w: 0.9 $c_1$: 2 $c_2$: 2 |
| FA | $\alpha_0$: 0.5 $\beta_0$: 0.2 $\gamma_0$: 1 |
| GA | Cr: 1 Mr: 0.05 Elite: 2 $\eta$: 1 $\alpha$: $\sqrt{0.3}$ |
| ACO | $\alpha$: 1 Q: 1 Initial ph: 0.1 Evaporation rate: 0.049 |
| CS | Discovery rate: 0.25 $\alpha$: 1 $\lambda$: 1.5 |
| DE | F: 0.5 Cr: 0.1 |

Este estudio emplea como algoritmo de clasificación una máquina de vectores soporte, *Support Vector Classification* (SVC), integrada con cada método de optimización en un enfoque envolvente para la selección de características. La función objetivo, definida en la ecuación (1), es una combinación ponderada del error de clasificación ($Error_{Predic}$) y la proporción de características seleccionadas respecto al total ($Ratio_{Sel}$), donde $\lambda = 0,9$. Por lo tanto, el objetivo de la metaheurística es minimizar esta función, que valora principalmente la tasa de acierto del modelo con el subconjunto de características.

$$fitness(x) = \lambda \cdot Error_{Predic} + (1 - \lambda) \cdot Ratio_{Sel} \tag{1}$$

Los resultados experimentales mostrados para cada conjunto de datos son los obtenidos en el conjunto de prueba utilizando la mejor configuración encontrada por la metaheurística según la función objetivo anterior en el conjunto de entrenamiento. Para garantizar la solidez de los resultados obtenidos, se han realizado 10 ejecuciones distintas en cada conjunto de datos. Como hay distintos conjuntos de datos, con valores de acierto muy dispares, vamos a usar principalmente los *rankings* o clasificación promedio. Para cada conjunto de datos, se ordenan todos los algoritmos según su posición en el

---

[1]`https://github.com/Migue8gl/TFG-Wrapper-Based-Metaheuristics-Feature-Selection`

ranking, mejor cuanto menor, y, a continuación, se calcula la media en los diferentes conjuntos de datos. Esta es una estrategia recomendada en la literatura [LaTorre et al., 2021].

# 4 Comparativas entre los algoritmos bioinspirados para Selección de Características

Esta sección se centra en comparar el rendimiento de los algoritmos de optimización bioinspirados recientes seleccionados para este estudio. El objetivo es evaluar su eficacia, eficiencia y robustez. Para conseguirlo, evaluaremos principalmente su desempeño en cuanto a la función objetivo (Sección 4.1), pero también consideraremos por separado el acierto (Sección 4.2), el factor de reducción de características (Sección 4.3) y, finalmente, el tiempo de cálculo (Sección 4.4).

## 4.1 Comparando la función de *Objetivo*

La Figura 1(a) muestra la clasificación media obtenida por cada algoritmo según la función objetivo. La mejor clasificación media la obtiene BGWO, que pertenece a la categoría que hemos denominado metaheurísticas modernas. Sin embargo, los siguientes algoritmos con las mejores clasificaciones pertenecen todos a la categoría de metaheurísticas clásicas:, BPSO, BCS, GA y ACO. Podemos apreciar que la mayoría de las metaheurísticas modernas ocupan las últimas posiciones, y únicamente el ABC obtiene peores resultados.

Figure 1: Ranking por *Fitness*

(a) *Ranking* Promedio

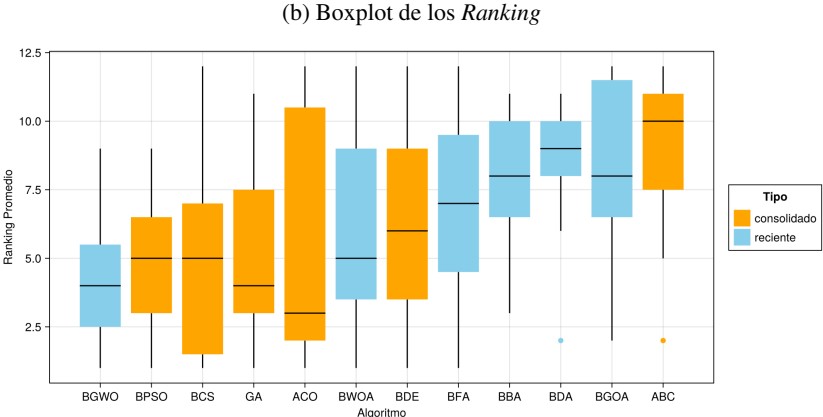

(b) Boxplot de los *Ranking*

Además, en la Figura 1(b) mostramos un *boxplot* con la distribución del *ranking* según la función *objetivo* de los distintos algoritmos. Es de destacar la robustez de los resultados tanto de BGWO como de BPSO. Por el contrario, aunque la mediana del ranking de ACO está significativamente por

debajo de estos dos algoritmos, muestra resultados mucho más irregulares en los diferentes conjuntos de datos.

Table 3: Resultados estadísticos en *fitness*: BGWO frente al resto de algoritmos

| BGWO vs | Original | Holm | Hommel | Hochberg |
|---------|----------|------|--------|----------|
| ABC | 0.001 | 0.007 | 0.007 | 0.007 |
| BBA | 0.004 | 0.043 | 0.034 | 0.043 |
| BDA | 0.005 | 0.048 | 0.043 | 0.043 |
| BGOA | 0.005 | 0.048 | 0.043 | 0.043 |
| BWOA | 0.073 | 0.511 | 0.406 | 0.511 |
| BDE | 0.107 | 0.642 | 0.535 | 0.599 |
| BFA | 0.135 | 0.677 | 0.599 | 0.599 |
| GA | 0.277 | 1.000 | 0.599 | 0.599 |
| BCS | 0.389 | 1.000 | 0.599 | 0.599 |
| ACO | 0.489 | 1.000 | 0.599 | 0.599 |
| BPSO | 0.599 | 1.000 | 0.599 | 0.599 |

Para garantizar que las diferencias observadas son estadísticamente significativas utilizamos tests estadísticos. En la Tabla 3 aplicamos los tests *post-hoc* no paramétricos de Holm, Hommel y Hochberg para controlar la tasa de error, siguiendo la recomendación de [Derrac et al., 2011]. Las pruebas revelan una mejora significativa de BGWO en comparación con cuatro algoritmos, el ABC y tres algoritmos recientes. Sin embargo, no se encontraron diferencias estadísticamente significativas entre BGWO y los demás algoritmos, incluidos los algoritmos establecidos BDE, GA, BCS, ACO y BPSO. Por tanto, no puede afirmarse que BGWO mejore significativamente a ninguno de ellos.

## 4.2 Comparando la Tasa de Acierto

En esta sección vamos a analizar el acierto medio obtenido por cada uno de los algoritmos por separado, que era el componente que tenía mayor peso dentro de la función objetivo.

En la Tabla 4, mostramos el acierto medio de cada algoritmo. Cabe destacar que no existen grandes diferencias de acierto entre la mayoría de los algoritmos, especialmente entre los 7 primeros. Es destacable el peor rendimiento en esta métrica de los dos últimos algoritmos, BGOA y ACO. BPSO y BGWO, que fueron los dos mejores en la función *objetivo*, obtienen resultados casi idénticos en términos de acierto. Por tanto, en la siguiente sección analizaremos el ratio de reducción conseguido por cada uno de los algoritmos.

Table 4: Acierto de cada Algoritmo

| # | Algoritmo | Acierto |
|---|-----------|---------|
| 1 | BCS | 0.774 |
| 2 | BFA | 0.770 |
| 3 | BGWO | 0.770 |
| 4 | BDE | 0.769 |
| 5 | BPSO | 0.767 |
| 6 | BWOA | 0.766 |
| 7 | ABC | 0.766 |
| 8 | BBA | 0.758 |
| 9 | GA | 0.754 |
| 10 | BDA | 0.749 |
| 11 | ACO | 0.729 |
| 12 | BGOA | 0.729 |

## 4.3 Comparando el Grado de Reducción

En la Tabla 5 se muestra la proporción media de características seleccionadas por cada algoritmo. Podemos destacar que existen grandes diferencias en el ratio de características seleccionadas entre los distintos algoritmos. ACO destaca con sólo un 14% de características seleccionadas, mostrando

una importante capacidad para reducir el número de características, aunque a costa de obtener un menor acierto. BPSO y BGWO tienen una menor capacidad de reducción, pero consiguen mejores resultados globales al obtener mayor acierto. En cuanto a la capacidad de reducción, destaca el buen rendimiento de un algoritmo consolidado como GA, al mismo nivel que BPSO y BGWO.

En cuanto a las diferencias entre algoritmos clásicos y recientes, podemos destacar que entre los 5 primeros algoritmos en términos de reducción, sólo BGWO aparece como representante de los nuevos modelos metaheurísticos.

Table 5: Proporción de características seleccionadas por algoritmo

| # | Algoritmo | Ratio |
|---|-----------|-------|
| 1 | ACO | 0.14 |
| 2 | BGWO | 0.34 |
| 3 | GA | 0.34 |
| 4 | BPSO | 0.36 |
| 5 | BCS | 0.47 |
| 6 | BDA | 0.48 |
| 7 | BWOA | 0.49 |
| 8 | BDE | 0.52 |
| 9 | BBA | 0.56 |
| 10 | BFA | 0.56 |
| 11 | BGOA | 0.57 |
| 12 | ABC | 0.77 |

## 4.4 Comparando el Coste Computacional

Dado que todos los algoritmos se han comparado utilizando el mismo número de evaluaciones como criterio de parada, podemos comparar el rendimiento de los algoritmos en términos de tiempo de ejecución. En la Tabla 6 se observa el tiempo medio de cada uno de los algoritmos en todos los conjuntos de datos. Se observa que existen diferencias importantes en los tiempos de ejecución. Si nos centramos en los mejores algoritmos en términos de valor de la función *objetivo* vistos anteriormente, podemos destacar que BGWO, que fue el algoritmo con mejor *ranking* promedio, tiene unos tiempos de ejecución bastante elevados en comparación con el resto. Sin embargo, BPSO, que alcanzó la segunda posición en el *ranking*, lo hace de una forma más eficiente. BFA es el algoritmo más rápido de los estudiados, pero no destaca por sus buenos resultados, como vimos en el apartado anterior.

Table 6: Tiempo Computacional Medio por Algoritmo

| # | Algoritmo | Tiempo (en segundos) |
|---|-----------|----------------------|
| 1 | BFA | 171.4 |
| 2 | BPSO | 881.6 |
| 3 | BDE | 953.5 |
| 4 | BWOA | 997.2 |
| 5 | BDA | 1098.2 |
| 6 | BGOA | 1098.9 |
| 7 | BCS | 1358.5 |
| 8 | GA | 1574.4 |
| 9 | BBA | 1991.1 |
| 10 | ACO | 2279.9 |
| 11 | BGWO | 3925.9 |
| 12 | ABC | 9525.0 |

## 5 Conclusiones

En los últimos años se ha propuesto un número significativo de algoritmos bioinspirados. Sin embargo, en algunos casos se ha cuestionado su valor innovador y su rigor experimental. Aunque muchos de estos algoritmos se diseñaron inicialmente para abordar problemas de optimización continua, también han surgido versiones binarias, sobre todo para la selección de características. Sin embargo,

las dudas sobre la eficacia de estos algoritmos se extienden también a sus homólogos binarios. Esto plantea una importante cuestión de investigación: *¿Superan los algoritmos binarios bioinspirados a los algoritmos binarios de selección de características más consolidados?*

Para responder a esta pregunta, hemos evaluado doce algoritmos, incluidos métodos modernos de bioinspiración y metaheurísticas consolidadas en la literatura, comparando su rendimiento en conjuntos de datos de clasificación idénticos. Cada algoritmo se aplicó para realizar la selección de características antes de utilizar el algoritmo de clasificación SVC. Hemos empleado una función objetivo que tiene en cuenta tanto el acierto como el porcentaje de características seleccionadas, y hemos analizado tanto los resultados en la función *objetivo*, como los ratios de acierto, los porcentajes de reducción de características y los tiempos computacionales.

El análisis revela que, en la mayoría de los casos, los algoritmos bioinspirados recientes comparados no mejoran significativamente algoritmos anteriores como los algoritmos genéticos (GA) o la optimización binaria por enjambre de partículas (PSO binaria). BGWO constituye una excepción, ya que ofrece algunas mejoras, aunque a costa de una mayor complejidad computacional. Así pues, nuestros resultados refuerzan el escepticismo sobre las ventajas prácticas de estos nuevos algoritmos, ya que sus innovadoras metáforas no parecen aportar una mejora sustancial del rendimiento. Creemos que este estudio proporciona una evaluación imparcial de la verdadera utilidad de estas nuevas propuestas bioinspiradas. Esperamos que sirva de guía a los investigadores a la hora de tomar decisiones fundamentadas sobre las técnicas de selección de características, contribuyendo así a un progreso más eficaz en este campo, valorando más la capacidad de los algoritmos que su popularidad.

## 6 Agradecimientos

Este trabajo está apoyado por los Proyectos de Generación de Conocimiento PID2023-149128NB-I00 y PID2023-150070NB-I00, financiados por el Ministerio de Ciencia, Innovación y Universidades de España.

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
