# OpenReview forum: "¿Son adecuados los Algoritmos Bio-Inspirados Binarios Recientes para Selección de Características?"
_MAEB/2025/Congreso — MAEB 2025_

### Official Review · Reviewer_tTMv · 2025-03-14
**¿Son adecuados los Algoritmos Bio-Inspirados Binarios Recientes para Selección de Características?**

**Rating:** 5
**Confidence:** 5

**Review:**

El artículo presenta una comparativa experimental de técnicas metaheurísticas recientes contra otras más tradicionales ya consolidadas en el campo para la reducción de conjuntos de datos para la posterior aplicación de técnicas de Machine Learning. Este problema en cuestión se denomina la Selección de Características y es uno de los pasos más relevantes dentro de un flujo de trabajo de ML. Dentro del campo de investigación predominan las técnicas como Principal Component Analysis (PCA) o Uniform Manifold Approximation and Projection for Dimension Reduction (UMAP) aunque las metaheurísticas evolutivas están ganando relevancia.

Los autores enmarcan el trabajo en una tendencia creciente dentro del campo, y es que cada vez se publican nuevos algoritmos bioinspirados con nombres cada vez más llamativos pero que en el fondo se basan en fundamentos ya previamente definidos en algoritmos tradicionales como el Particle Swarm Optimisation, Genetic Algorithm o Differential Evolution. Considero que el trabajo debe ser aceptado para su publicación en el congreso basandome en:

- El trabajo está muy bien estructurado y justifica debidamente la relevancia de la cuestión de fondo que se aborda (evitar despreciar algoritmos tradicionales por alternativas modernas sin una evaluación contrastada).
- El problema definido para evaluar las técnicas modernas y tradicionales es de gran relevancia y actualidad dentro del campo.
- Los algoritmos elegidos para la evaluación, así como los conjuntos de datos y el diseño de la evaluación experimental permiten validar suficientemente las conclusiones obtenidas por los autores.

Los resultados obtenidos muestran que solo una de las técnicas modernas evaluadas (BGWO) presenta algunas mejoras aunque se aprecia una gran carga computacional añadida. Me gustaría poder acceder al repositorio mencionar en el trabajo ya que aunque la complejidad computacional medida en tiempo de ejecución (segundos en este caso) depende en gran medida del ecosistema en el que se ejecuta y la propia implementación del algoritmo, los resultados muestran una diferencia bastante significativa. Creo que este tipo de trabajos son muy importantes para la mejora del campo y poder evitar caer en la tendencia de apreciar las nuevas técnicas simplemente por su novedad sin tener una visión crítica de lo que realmente aportan al campo.

Finalmente,  me surge una pequeña duda con la definición del problema de Selección de Características (línea 31). En particular, creo que la definición está redactada al revés del consenso dentro del campo y es que se suele entender por SC como la reducción de los conjuntos de datos para seleccionar aquellas características más relevantes y que mejor representar las instancias del mismo. En este caso, los autores proporcionan una definición inversa en la que dan a entender que el problema de la SC se centra en seleccionar aquellas que van a ser descartadas.

---

### Official Review · Reviewer_dud9 · 2025-03-17
**Son adecuados los algoritmos Bio-Inspirados binarios recientes para selección de características.**

**Rating:** 2
**Confidence:** 4

**Review:**

Este artículo presenta un estudio comparativo entre algoritmos bio-inspirados binarios recientes y metaheurísticas clásicas para la selección de características en problemas de clasificación. La principal conclusión es que los algoritmos modernos no ofrecen mejoras significativas sobre los tradicionales, y cuando lo hacen, es a costa de un mayor coste computacional. Sin embargo, este resultado no es sorprendente, por lo menos para el tipo de problema planteado. Estoy totalmente de acuerdo con los resultados de los autores, pero la experiencia demuestra que los algoritmos diseñados específicamente para un problema suelen superar a enfoques generales. Además, desde un punto de vista teórico, los teoremas de *no free lunch* sugieren que ninguna metaheurística es universalmente mejor, lo que implicaría la necesidad de comparar "todos" los algoritmos bio-inspirados con "todos" los métodos clásicos, algo muy dificil de realizar.

El trabajo es meteorológicamente sólido y presenta una evaluación experimental, pero su contribución al conocimiento es limitada, ya que confirma un hecho mas o menos conocido en la comunidad. Experimentalmente es dificil de validar la pregunta planteada. Sin embargo, algunas intuiciones sobre este tipo de algoritmos podria dar lugar a algunos aspectos formales que den luz al por qué este tipo de algoritmos podrían ser "los mismos" pero "disfrazados" de cierto folklore bio-inspirado.

---

### Official Review · Reviewer_ZvEu · 2025-03-17
**La experimentación debe incluir conjuntos de datos y algoritmos estado del arte para el problema propuesto**

**Rating:** 2
**Confidence:** 5

**Review:**

Este trabajo abunda en la opinión cada vez más generalizada sobre la no necesidad de proponer más algoritmos bioninspirados que en realidad podrían verse como casos particulares (y rebuscados) de algoritmos bioinspirados clásicos, requiriendo además parámetros adicionales que hay que optimizar. El trabajo se centra en "comprobar" esta tendencia de no mejora significativa en el problema concreto de la selección de variables. El resultado obtenido es la confirmación de la hipótesis formulada: efectivamente no hay mejora significativa.

Se trata de un trabajo puramente experimental y por ello la experimentación debe ser rigurosa. En este sentido no basta con comparar los algoritmos estudiados entre sí o frente a algoritmos estándar de computación evolutiva. Al tratarse de un problema estándar y muy estudiado en aprendizaje automático, se deben incluir en la comparación técnicas estándar de selección de variables no evolutivas. Por otra parte, la selección de los conjuntos de datos es cuestionable, ya que para un artículo de selección de variables, los conjuntos de datos seleccionados tienen un número pequeño de variables, entre 4 y 60, lo que está completamente fuera del estándar actual de la literatura en selección de variables.

---

### Decision · Program_Chairs · 2025-03-20

Accept